# Smart Thinking on Co-Creation and Engagement: Searchlight on Underground Built Heritage

**Carlos Smaniotto Costa** [1,*] , **Rolando Volzone** [2] , **Tatiana Ruchinskaya** [3] , **Maria del Carmen Solano Báez** [4] , **Marluci Menezes** [5] , **Müge Akkar Ercan** [6] **and Annalisa Rollandi** [7]

1   Department of Architecture and Urban Planning, Universidade Lusófona, Campo Grande, 376, 1749-024 Lisbon, Portugal
2   Department of Architecture and Urbanism, ISCTE-University Institute of Lisbon, Av. das Forças Armadas, 1649-026 Lisbon, Portugal
3   TVR DESIGN CONSULTANCY, 6 Green Street, Willingham, Cambridgeshire CB24 5JA, UK
4   Department of Applied Economics, University of Murcia, Campus de Espinardo, 30100 Murcia, Spain
5   Laboratório Nacional de Engenharia Civil, Av. Brasil, 101, 1700-066 Lisboa, Portugal
6   Faculty of Architecture, Department of City and Regional Planning, Middle East Technical University, Dumlupınar Bulvarı, 1, Ankara 06800, Turkey
7   Department of Environment Constructions and Design, Institute of Earth Sciences, University of Applied Sciences and Arts of Southern Switzerland, Via Flora Ruchat-Roncati, 15, 6850 Mendrisio, Switzerland
*   Correspondence: smaniotto.costa@ulusofona.pt

**Abstract:** This paper aims to explore public participation for activating underground built heritage (UBH). It describes and analyses practices of stakeholders' engagement in different UBH assets, based on experiences gathered in the scope of the European COST Action 'Underground4value'. It brings together five inspiring cases from Italy, Portugal, Spain, Switzerland and the United Kingdom, in which digital and mobile technologies were used as tools to improve community experiences in UBH. Thus, the paper discusses 'smartness' from the perspective of people and communities around cultural assets, where 'smartness' becomes a new connotation and a pathway to advance (local) knowledge and know-how. Therefore, this paper takes on the challenge to define a smart city as an ecosystem for people's empowerment and participation, and, in particular, to explore social tools for creating new values in heritage placemaking—where sharing knowledge becomes a fundamental principle.

**Keywords:** underground built heritage; citizens engagement; co-creation activities; smart participatory processes

## 1. Introduction

"Underground Built Heritage (UBH) is represented by three types of building activities, namely architectural, urban and landscape heritage beneath the surface of the earth" [1] to which the contemporary generation attaches cultural values. This definition is adopted by the European COST Action 'Underground4Value', which aims at establishing an expert network for promoting balanced and sustainable approaches to preserve the UBH and to realise the potential of underground space in urban areas for regeneration policies [2]. The term UBH refers thus to the rich diversity of manmade inherited underground assets, such as houses, cisterns, burial places, galleries, etc. Some examples are explored in the outcomes of this Action [3–5]. The Action 'Underground4Value' also aims at providing a better understanding of UBH as a catalyst for sustainable local development, which calls for community engagement. As Maksymiuk et al. [4] rightly pose, activating a UBH asset starts by thinking from the depth and ends by acting on the surface. The interaction between the beneath and the surface can be the anchor for responsive heritage awareness and smart action in cities. In this activating process, one has to consider functional, social and cultural factors along the legislative and regulatory frameworks [3].

The pandemic and climate crises have underlined the importance of the cultural heritage assets and public spaces in everyday life because they have social and cultural connotations and reflect the identity of the local communities. As Gehl argues [6], public spaces manifest in the physical world, but what really makes them tick is people. The current discussion on public space includes two main issues, which should be taken into account during its design. It is the resilience of places and a holistic approach to gain a comprehensive vision of the problems and needs of the local community. The sudden changes during the COVID-19 emergency have emphasised the importance of knowing how spaces can be adapted to support societal needs. Climate change is another crucial societal challenge that should be considered in the design of public places and in the preservation of UBH. Over the last few years, its effects have been increasingly evident and have repercussions on daily life. Cities cannot miss the opportunity to guide the transition, having greater awareness of their characteristics and the potential risks to which they are exposed. 'Underground4Value' understands that activating and promoting a UBH is a transition process, from an undervalued/forgotten asset underneath the surface to one, which is centred in the core of the community. Working with the community provides a positive effect and helps harness the power of communities. 'Underground4Value' thus sets a fundamental milestone for integrating underground assets, landscape surfaces, cultural heritage studies, and governance. For its nature, the planning for UBH activation should be both government-fed and community-based. Such a combination is what makes the activation a smart process.

A smart city is more than a technology-embedded city. We embrace the European Commission's view and definition of a smart city, which is a "place where traditional networks and services are made more efficient with the use of digital solutions for the benefit of its inhabitants and business" [7]. This understanding encompasses three central issues: the place, the (traditional) networks, and the benefits to be gained through the use of new technologies. In other words, a smart city goes beyond the simple addition of digital technologies; it involves networks to achieve a better quality of life. As de Lange [8] argues, the networked city should encourage encounters and deal with the unexpected and the differences. The author further claims that a technology-driven development does not per se empower citizens to become active 'players' and 'hackers' of their own cities. Moreover, he warns that technological fixes cannot solve complex urban problems. Sennet's point of view [9] allows us to reflect on ICT as a resource for a smart city to increase people's cognitive capacity to reason and understand their own territory. In this sense, a smart city is an essential contribution to move forward in the co-creation of an open city which is controlled and shared by people. This new perspective poses the question: can smart cities be achieved without community engagement and local knowledge? An answer to this question particularly necessary when the issue is tackling underground assets, which have distinct values and attributes, primarily discussed within the COST Action 'Underground4value' [2,3,10].

The overarching aim of this paper is to analyse the community engagement and co-creation in activating UBH assets, as an added value for a smart development. With this end in view, it discusses the engagement practices on the basis of five inspiring cases from Italy, Portugal, Spain, Switzerland and the UK. By studying the experiences and practices of stakeholders' engagement in these different countries and investigating their strategies to raise awareness of UBH, this paper ultimately seeks to discuss 'smartness' from the perspective of people and communities around this critical cultural asset.

## 2. Materials and Methods

Five examples of citizens' engagement with assets of UBH and the use of digital technologies were taken as paradigmatic cases to discuss that smart digitisation can be a useful tool for empowering people and to engage them in the co-creation and preservation of their own culture. The cases included the study of digital media platforms in the Italian city of Matera (*Matera Città Narrata*—Tales of a City) and the Escoural Cave in Montemor-o-

Novo (Portugal), the identification and classification of the heritage linked to the Spanish Civil War in Valencia (Spain), the COMEVA(?) project in the Mendrisiotto Region (Canton Ticino, Switzerland) and the A14 road improvement scheme in Cambridge (UK). These cases were analysed within 'Underground4value' to gain knowledge and experiences on informal planning approaches and participatory processes. Variegated data was gathered from a variety of sources. A literature review provided important insights to base the drafting of the lessons from these cases.

The methodological approach was structured in two steps. Step 1 consisted of bibliographic and documentary research on the cases in order to identify the main achievements of each case, considering their historical, political, social and urban contexts. The second step focused on a comparative analysis of cases to identify their similarities and differences and to draw lessons. This cross-case analysis was used as a research method, as it can mobilise knowledge from individual case studies [11], and supports the objective of demonstrating the similarities and differences between the cases. From the cross-case analysis, key lessons and features of civic engagement were identified and discussed. The efforts in the five cases in activating UBH assets were distilled into four key lessons (discussed and justified in topic 4).

## 3. Analysed Experiences

### 3.1. Matera Città Narrata—Tales of a City, Matera, Italy

Matera is one of the most complete surviving rock-cut settlements in the Mediterranean region. Matera, located in the Basilicata region of southern Italy, is an extraordinary UBH settlement with its uninterrupted continuity of human presence from 12,000 years ago until today; since 1993 it is included in the UNESCO World Heritage List [12]. Thousands of natural caves, which allowed the inhabitants to be settled and protected, are the prominent characteristics of Matera. The city gained international fame with Sassi, its ancient urban centre, which accommodates houses dug into the calcareous rocks and the streets laid on the rooftops of these houses. Due to the dire living conditions, in the 1950s the Italian government forcefully relocated most of Sassi's population to the new area of the city. Until the late-1980s, Sassi was considered a poverty site since these houses were mostly unliveable. However, the municipality of Matera developed a tourism-oriented regeneration policy for Sassi with the aid of the European Union, the government, UNESCO, and creative industries. Many parts of Sassi were restored and re-occupied for tourists, craft activities or living [13,14].

*Matera Città Narrata* (Tales of a City) is one of the projects behind the successful heritage-led regeneration of Sassi. It is a digital media platform (www.materacittanarrata.it, accessed on 14 October 2022) launched in 2012 for Matera's candidacy for the European Capital of Culture 2019 [15,16]. It provides information for planning a visit and finding one's way in Sassi and its surroundings. The most appealing content for online visitors is the contributions and engagement of local people who tell their stories about what it means to grow up and live in this unique settlement [17]. *Matera Città Narrata* was developed in cooperation with the National Research Council (CNR), The Institute for Technologies Applied to Cultural Heritage, and creative Web and multimedia businesses: Imagimotion, Net Agency/Netway, HSH Informatica e Cultura and Dinamo Italia. The Giuseppe Buonsanti Archive provided historical photographs; and artists and residents of Matera told their stories and views about the Sassi. The Regional Promotion Agency (Agenzia di Promozione Territoriale di Basilicata—APT Basilicata) and the Basilicata Regional Government/Department for Production Activities financed the project. In November 2013, the selection panel evaluated applications from Italian cities and five other cities, and Matera was short-listed.

This media-rich platform introduces domestic and international tourists to Matera's cultural landscape, history and intangible heritage. In this way, it aims at inviting tourists to stay and experience the city for longer than just a few hours, potentially increasing tourism income. The content of *Matera Città Narrata* is provided in different formats, including

84 videos, 67 soundtracks, 30 slide shows and other multimedia content, and itineraries and excursion plans. There are also six interactive virtual reality panoramas for "flying" over the Matera area, based on 3D reconstructions of different historical development stages of the site, such as the Neolithic age, the classical period, and the late 19th century. A platform, developed through the residents' engagement, presents Matera's authentic living heritage by offering residents' specific highlights, stories and visions in their own words with 2- or 3-min videos. Historical photographs on the website add much to the understanding of the context. The platform offers free web and mobile access to a vast amount of information on the heritage values of Matera. It also enables the engagement of further stakeholders by accepting contributions from institutions, businesses and other residents (e.g., school classes), providing a section where they can share their own content. Contrarily, tourists are not allowed to express their opinions on the website.

The content of *Matera Città Narrata* can be accessed in different ways and formats, according to the device and preference of the user (i.e., smartphone or an older mobile phone, iPad, PC). The multimedia and other advanced content can be explored on the website and downloaded by iPad users in Italian, English and German. Furthermore, various applications for mobile devices with different operating systems (iOS, Android) are offered. For example, itineraries, excursion plans and multimedia content about particular places can be downloaded. Free Wi-Fi access has been implemented in the Sassi neighbourhood, so visitors with smartphones can access the website or another interactive application. A mobile service is also available for visitors without smartphones or connections. They can call a mobile service that sends multimedia messages or allows listening to audio-guide content about sites and monuments in Matera (where a sign with an ID number has been placed). Additionally, a treasure hunt application for iPhone is offered for children.

To sum up, *Matera Città Narrata* is an example of the successful cooperation of organisations from different sectors, including several centres of the National Research Council (the one responsible for the overall concept and coordination), the municipality, regional agencies, heritage institutions, and media companies. Regarding the UBH valorisation, a critical feature of the project is the engagement of people living in Matera, which contributes much to the spirit and outcomes of the project. In addition, it achieves a good balance of research-based content (e.g., virtual reconstructions) and the human side—voices, knowledge and authentic experiences of scholars, artists and children. Regarding the technology and innovation aspects, *Matera Città Narrata* offers several ways of accessing a variety of content depending on the type of device used by users. The main focus of the project is not on the new technologies, but on access and storytelling. Nonetheless, depending on the capability of the user device, richer forms and interactivity of the content are provided.

### 3.2. Escoural Cave, Montemor-o-Novo, Portugal

Montemor-o-Novo is located in the Alentejo region, about 30 km west of the city of Évora, and has been listed as a UNESCO World Heritage Site since 1986. The vast cultural heritage of Montemor-o-Novo reveals the millennial occupation of the territory. It is extremely rich in terms of natural and cultural heritage resources, with caves, megalithic remains (erected between the 6th and 3rd millennium B.C)., a castle, convents, churches, chapels, and among these, the oldest one: the UBH settlement Escoural Cave (Gruta do Escoural). This cave is about 15 km south of Montemor-o-Novo historical centre and consists of multiple galleries, with rock art paintings and engravings. Here, an important archaeological collection was discovered, comparable to the caves in Altamira and Lascaux. Neanderthal nomads took shelter here when hunting. During the Upper Palaeolithic (around 35,000 years ago), it was used as a sacred space: rock art paintings and engravings belong to this period. In the Neolithic (7000 years ago), it was adapted to a necropolis. This was, apparently, the last use of the cave. Indeed, its entrance was blocked, probably intentionally or due to natural processes.

In 1963, during marble quarrying works, an entrance was accidentally rediscovered. For the first time, in Portugal, traces of Palaeolithic rock art (around 50,000 years ago)

were identified. Very quickly surveys and investigation of bones and archaeological pieces and excavations took place. In 1965, the interior rock art was surveyed, too. In this same year, the Escoural Cave was declared a monument of national interest. Between 2009 and 2011, the renovation of the cave's surrounding area and the rehabilitation of the visitor's facilities were carried out. In these years, for the first time, a 3D survey with laser scanning techniques was produced and a decade later a new 3D survey, integrating laser scanning and aerial photogrammetry mapped the cave, its interior and exterior. A total of 73 virtual models were created and made available online in the open access platform Sketchfab (https://sketchfab.com/search?q=escoural+cave&type=models, accessed on 30 October 2022), reaching millions of users. Two models are of the interior and exterior survey; the remaining 71 represent a vast selection of artefacts: pottery, cups, awls, arrowheads, rock art engravings and cave paintings. These can be visualised online at 360 degrees with historical information or can be downloaded on personal devices.

Thanks to a partnership between the Montemor-o-Novo municipality and Sketchfab, these 3D models can be directly visualised on the online platform Morbase (https://montemorbase.com, accessed on 30 October 2022). Morbase aims to promote scientific knowledge about local cultural heritage both for the general public and the scientific community. In this way, awareness of nearby communities and public administrations can be raised. Moreover, new initiatives for heritage conservation, valorisation and rehabilitation are encouraged, stimulating interest in visiting these spaces [18]. This platform has been created for and by the locals. Indeed, they have an active role, being integrated into the process of creation and knowledge dissemination. Beyond the local community, the platform also reaches a wider audience: regional, national and international. Within this open-source platform, heterogeneous contents are integrated and displayed: a digital library with open access scientific articles about the Montemor-o-Novo; an interactive map with the identification of the local historical and cultural heritage; documentaries about local assets; virtual tours; a virtual museum, with 85 interactive 3D models—obtained by 3D surveys or digital reconstructions—including archaeological sites, built heritage, decorative details, small artefacts, among others; a mini-series on local know-how, that is slowly being lost, as well as videos promoting local gastronomy. This is the rigorous result of an effort obtained through the integration of knowledge-based documentation methodologies and digital information for the study of historical and cultural heritage [17]. Moreover, 3D printings, as demonstrated in the scientific research [19], can enable the reconstruction of high-resolution physical replicas. Currently, some virtualised artefacts from Escoural Cave have been printed, such as stone knives, clay pots and human skulls; this paves the way for the printing of large size objects, as was the case for the Chinese Yungang Grottoes [20] and in Europe, for Altamira (Spain), Lascaux II, Lascaux IV and Chauvet II (France). Recent studies reveal a good reception by visitors, through an authentic heritage experience, when examining these European cases [21], as well as a specific study about the cave of Chauvet [22]. The issue of growing visitors needs to be discussed with increasing urgency, in order to safeguard the more fragile sites from external factors, such as over-tourism, which can compromise the integrity of these UBH sites.

Since 2020, the FIRST-ART project (https://first-art.org, accessed on 30 October 2022) is carrying out a joint investigation and promotion of the Escoural and the Spanish Maltravieso caves. New methodologies are being put in place, such as digital image processing, high resolution 3D scanning of the rock art and integral 3D modelling of the cavity, pigment and binder composition analysis, absolute chronological dating and identification of DNA in the pigment composition. At the same time, an updated catalogue of the rock art of the Escoural cave is drawn up, and a permanent exhibition for the Information Centre of the Escoural Cave is being produced. This includes new content, such as interviews with people involved in the discovery of the cave; among them is the first worker who entered the cave, an 18-year-old at that time. The local community is engaged in different stages. A multilevel engagement strategy is carried out, in order to collect memories and to connect people with UBH, raising awareness about the local heritage. First, the community

was involved in the collection of records from local, regional and national public archives, including documents, photos, newspaper reports and television documentaries about the discovery, and archaeological works. The community was also involved in the processing of these documents and their adaptation to tourism purposes. The Escoural Parish Council has been engaged in the whole process. Indeed, for example, the visits to the cave are guided by a local resident, following an agreement between the Regional Directorate of Culture, the City Council of Montemor-o-Novo, the Parish Council of Escoural and a local association.

In 2018 and 2019, the Escoural Cave received about 3500 visitors per year, with an average of 14–15 people per day. Taking into account the geographical location, the opening days and the daily visitor limit of 40 people, this average can be considered as a growing interest in the cave. In the years 2020 and 2021, the average number of visitors fell due to pandemic restrictions and by the specific features of this UBH site. However, thanks to the efforts of the municipality in the digitisation of cultural heritage, these values have been supplemented by virtual visits.

*3.3. Spanish Civil War, Valencia, Spain*

The Regional Administration of Valencia (RAV, Generalitat Valenciana) made a great effort to identify, catalogue, and classify the heritage linked to the Spanish Civil War. The Spanish case brings to the table a continuous experience of the involvement of different key agents, particularly citizens, through actions to promote participation, and the implementation of participatory processes. It provides a brief description of the main milestones that have made it possible to initiate the participatory valorisation process of a vast heritage that is not only beneath the surface, but also beneath the collective consciousness. This experience can be systematised by two key drivers and their actions: the first, an initial impulse from the RAV, through an inventory of the heritage (identifying, cataloguing and disseminating) to give rise to feedback from the population and extend the documentation process; and the second, an articulation process of the different agents to enhance the value of the heritage through information and awareness-raising actions to stimulate engagement and participatory decision-making processes for the recovery and conservation of heritage and historical memory.

The cataloguing process began in 2017, following the approval of the law on the *Patrimonio Cultural Valenciano* [23], which granted the consideration of immovable assets of local relevance to the historical and archaeological civil and military heritage of the Civil War in the Valencian Community. These include, for example, the buildings that served as the headquarters of the government of the Republic, as well as the relevant spaces used during the war period between 1936 and 1939 and built before 1940. This fact led to the participatory preparation of a specific inventory of these assets by the Regional Ministry of Education, Research, Culture and Sport (the Regional Ministry responsible for the matter) in collaboration with the Valencian Cartographic Institute, the Polytechnic University of Valencia, and the Directorate General of Information and Communication Technologies of the Regional Ministry of Finance and Economic Model. The objective was to contribute to the recovery of heritage assets with high value in terms of military architecture, especially to promote knowledge and respect for all of the Civil War's remains and its memory due to their historical and cultural value. The result of the compilation process is an interactive inventory that presents a structured classification of the wartime heritage of the Civil War. This inventory brings together unprecedented information that was scattered, and cataloguing that was missing. The elaboration lasted more than a year and a half, and it was carried out through on-site visits to different parts of the Autonomous Region of Valencia. This process was complemented with a co-creation strategy that consisted of sending and receiving requests for information to different levels of the administration (provincial and local), along with other public and private agents (universities, civic and cultural associations, etc.), enabling their involvement throughout the process [24]. A website was created in January 2019 (www.patrimonigc.gva.es, accessed on 20 October 2022),

which represents the beginning of a shared and open inventory. One of its most notable features is the possibility of expanding and specifying the information, supported by the participatory dimension, which pursues the permanent coordination and engagement of all the aforementioned agents [25]. The aim is to facilitate the access and comparison of all this information centralised in the virtual space so that they can contribute to the cataloguing process. Thus, the inventory offers an extensive list of heritage elements, particularly UBH related to the military architecture of the Civil War, classified into eight categories (Cultural and Political Centres, Active Defence, Passive Defence, Detention Spaces, Weapons, and Ammunition Factories, Logistical Infrastructures, Civil Works, Civil and Military Health, Detention Spaces, Weapons, and Ammunition Factories, Logistical Infrastructures, Civil Works, Civil, and Military Health) and 54 typologies (Air Raid Shelters, Fortifications, Trenches, Bunkers, Ditches, Shooting Pits, Resting Constructions and Caves, Hiding Places, to name a few).

The inventory offers a diverse sample in terms of typology and territorial distribution made up of 54 assets, 18 for each of the three provinces, for which sufficient information is available and systematised in a standardised way, namely: location, georeferencing, typology, inventory status, description, graphic documentation, and sources of information. Throughout the cataloguing process, it was also verified whether or not each of the 54 assets had been included in the General Inventory of Valencian Cultural Heritage. In addition to the sample, 563 heritage assets were identified, for which their basic data are available: location, georeferenceing, and typology. Thus, mapped information is offered for 617 heritage assets from the Civil War, 315 from Castellón, 174 from Valencia, and 128 from Alicante. There were also incorporated, without homogenising its content, 453 files from the province of Alicante, 433 from Castellón, and 180 from Valencia carried out by three researchers (one for each of the provinces) of the Regional Ministry of Education, Research, Culture and Sport compiled during 2017.

The information and awareness-raising actions to stimulate engagement and participatory decision-making processes linked to the recovery of war heritage are of a continuous and transversal nature, with the agents involved determining the steps to be followed in the assessment process. The 2017 regulatory update was accompanied by a call for grants aimed at promoting cultural heritage [26]. These funding grants have been used to finance archaeological, architectural, historical-artistic interventions, and other actions aimed at generating products based on intangible ethnographic elements. The beneficiaries of the funds are a wide variety of agents, including town councils, foundations, associations and non-profit organisations [27]. There are also public grants from the Regional Administration of Valencia to carry out actions related to the Valencian historical and democratic memory, the enhancement of places of the Memory, and the removal of vestiges related to the Civil War.

Among the actions for the recovery and enhancement of heritage, there are two worth mentioning:

- The XYZ Defensive Line, the largest defensive line in Spain, was built in 1938 with an immense number of trenches, bunkers and other underground constructions [28]. Presently, a project is aimed at transforming the fortified line into a tourist route connecting 40 towns in Castellón, under the leadership of the Agencia Valenciana de Turismo, Universitat Jaume I and the Federación Valenciana de Municipios y Provincias [29,30].
- The Camp d'Aviació de Vilafamés Project 442 (Vilafames Airfield) foresaw the conservation of the remaining airfield, and the publication of the book *El aeródromo militar de Vilafamés* by Carlos Mallench, Blas Vicente, Jose F. Albelda and Josep J. Mirallés, and the documentary *442: El camp d'Aviació de Vilafamés. História d'un aeródrom* [31,32]. This project involved the Town Council of Vilafamés through its Department of Tourism, the company Arqueocás, and historians. The local community was engaged in social archaeology activities, i.e., in the construction of historical knowledge through archaeological and/or historical tasks.

*3.4. COMEVA(?) Project, Mendrisiotto Region in Canton Ticino, Switzerland*

The COMEVA(?) Project (www.communityofvalue.com, accessed on 20 October 2022) is an incubator of ideas for public spaces in the Mendrisiotto Region. Its goal was to create and strengthen the community around territorial issues through social media. The analyses conducted during the last years underlined the absence of a specific vision for the Mendrisiotto Region. Therefore, in 2022, the project moved towards raising awareness necessary to understand the needs of the new generations. The strategy was developed in two phases. The first aimed at creating an online community around the Mendrisiotto, while the second collected users' ideas for the area. It will be possible to propose other discussion topics, one of them are certainly the UBH assets, e.g., the Cement Path. The COMEVA(?) aimed to engage young people between 15 and 35 years old, with particular attention to two specific segments: university students of SUPSI and those enrolled in high schools in the cross-border region between Switzerland and Italy (Gen Z). The goal was to collect ideas and suggestions for planning and management of open spaces through the Instagram account @comeva_community_of_value. The materials were mapped and made available on the official website. In that way, the project created the base for a bottom-up discussion that engages the new generations [33]. The work was divided into three exercises: two were exclusively online, while the third took place at the new SUPSI Campus in Mendrisio (CH). The first exercise took place online through a contest. The goal was to collect reports and suggestions for public spaces in the cross-border region. Users were free to highlight eye-catching projects and ideas to improve territorial sustainability or report public places to be redeveloped. All these contributions could be sent to the official Instagram account in three simple steps described online. The community freely chose the method of sharing: posts, stories or videos. The objective of the first exercise was to collect the materials and to create a map of ideas for the cross-border territory.

The second exercise also took place exclusively online. According to the current digital communication trends, the instruments that supported the test were the stories on the official Instagram account. The goal was to collect users' insights on the quality of public spaces. Thanks to this platform, the stories presented two alternative visions of public space and the community to indicate their preference with a simple click. The data was updated in real-time and visible for 24 h. The tool that supported the realisation of the image of public space was the parametric design. That instrument helped translate into the exact resolution, alternative solutions for some public spaces in the cross-border region. In this way, the community could compare different projects and their effect and then express a valuation. The application of parametric design in urban planning can be extensive. Following the project goal, the use of parametric design was to aid participatory planning. Lee asserts that decision-making in design is a cognitive process where alternatives are generated and evaluated, potentially enabling a more creative design process. In recent years, parametric design's heightened capacity for automatically generating and evaluating options has been celebrated by researchers and designers, but it has also placed an increased emphasis on decision-making activities [34]. The final third part of the research took place offline. The goal was to verify the impact of the offline events on topics and discussions that started online concerning the Mendrisiotto region. In particular, the project aimed to verify the importance of this area on social platforms. The main questions were: what is Mendrisio's image in social networks? Is there a virtual collective idea around Mendrisio?

Regarding the future image of the territory, it is helpful to start with the elements that distinguish the area and analyse the case studies. Mendrisiotto has excellent potential, but its visual identity online and offline has not been recognised. There are numerous caves dating back to 1915 and 1965. Some were used to extract materials; others were the headquarters of companies. In addition to the caves, there is an exciting hiking network and various parks. The Gole della Breggia Park (www.parcobreggia.ch, accessed on 20 October 2022) is a place that showcases the presence of UBH, local identity, and citizens' engagement activities while preserving the natural environment and providing environmental comfort, even in the hottest summers. The Park is located in the Muggio Valley, not far from the

national border with Italy [35]. The old Saceba building, a cement factory operated from 1963 to 1980, is located in the Park's centre. The company had one of the most prosperous economic activities in the area and finished its operations in 2003, five years after the Park was established. During the same year, Holcim Switzerland (SA) acquired Saceba. A few years later, in 2008, they renovated the remaining buildings and added them to a historical and environmental itinerary. One of the most representative assets is the 2 km Cement Path (www.percorsodelcemento.ch/it, accessed on 20 October 2022), which connects open spaces, extraction galleries, and the Furnace Tower, a space used for local events and exhibitions. Moreover, the Park has become a social place combining environment and collective memory.

Mendrisiotto Region has all the elements to create a strategy starting from the UBH. To complete a virtual process capable of attracting online and offline users, it seems helpful to start from the ideas of GenZ and analyse the activities implemented by The Gole della Breggia Park. COMEVA(?) is investigating the imaginary to understand if by arranging all the elements, it is possible to create a future vision.

*3.5. A14 Improvement Scheme, Cambridge, UK*

The A14 Cambridge to Huntingdon improvement scheme is an example of the Nationally Significant Infrastructure Project (2014–2019), with one of the largest archaeological programmes ever undertaken in the UK. The scheme involved a major upgrade of the 30 km (19 miles) length of the A14 highway between Cambridge and Huntingdon, including a new 12-mile bypass for Huntingdon, widening of the A14 and A1 roads, construction of 34 new bridges, and new and improved junctions [36]. The A14 Integrated Delivery Team for Highways England was a joint venture of Costain, Atkins Balfour Beatty international infrastructure group, Skanska and Jacobs. The project was awarded the Considerate Constructors Scheme's National Ultra Site Awards of 2019, the 2019 Ultra Site of the Year Award and the Archaeology Award 2019. Pre-application consultation is a statutory requirement for Development Consent Order (DCO) applications for large infrastructure projects under the Planning Act 2008, and associated guidance encourages additional informal non-statutory engagements [37].

The Planning Act 2008 sets out principal statutory consultation processes and requirements for pre-application consultation and types of consultees, including prescribed consultees (e.g., statutory organisations), local authorities, relevant land interest consultees, local community, people and organisations commenting on the proposals [38]. A two-stage public consultation for the project was undertaken in autumn 2013. The first stage was non-statutory and set out the initial route options and tolling proposals in a Consultation Report produced in December 2013. A statutory pre-application consultation was held between April and June 2014 [39]. Prior to this, the consultation was publicised within local and national papers. A draft of the Statement of Community Consultation (SoCC) with local authorities was prepared and published in spring 2014 before the statutory consultation started. The SoCC included details of the proposals, the planning process and how the Highways Agency proposed to consult with the local community. The Strategic Stakeholder Board was set up in 2014 to ensure stakeholders are appropriately engaged in detailed design issues. It included Cambridgeshire County Council, Greater Peterborough and Cambridge Local Enterprise Partnership, the Department for Transport, South Cambridgeshire District Council, Huntingdonshire District Council, Cambridge City Council, Network Rail and the Port of Felixstowe.

The stakeholders were engaged through a number of forums for strategic stakeholders, parish councils, communities, the environment and landowners. Community forums were used for each relevant community affected by the scheme for consultation by both Highways England (the client) and by the local authorities on detailed design. Highways England's Stakeholder Manager for each forum co-ordinated all stakeholder meetings. Three distinct consultation and engagement processes have taken place since April 2014 [38]:

- A statutory formal consultation period between April and June 2014;
- Statutory consultation with additional land interest consultees;
- Non-statutory design change engagement between September and October 2014.

The aim of the consultations was to provide a "design sympathetic to the needs and wishes of stakeholders, delivering an essential upgrade to the route in line with the Highways Agency's objectives" [38]. The Consultation Report was submitted to the Planning Inspectorate in December 2014 alongside the DCO application in accordance with Section 37 of the Planning Act 2008. It provided consultation responses to each element of the scheme and by key topics raised (e.g., traffic, noise) and reported feedback by consultees.

The Highways Agency used a wide variety of consultation tools to ensure everyone interested in the proposals had the opportunity to participate. These tools included a series of public exhibitions, leaflet drops, online information and media announcements. The consultation engaged with 1390 consultees, held 31 public consultation exhibitions, attended 187 meetings with local authorities and other stakeholder groups and held two webchats [38]. A questionnaire was used as one of the routes for the local community and general public to feedback their views on the proposal. The engagement activities promoted more than 1152 responses, with a total of 8350 comments. These comments were organised into 15 overarching themes, which were subdivided into 48 categories. Over four in ten respondents (46% to a maximum of 85%) agreed with the individual elements of the scheme, while under three in ten (8% to a maximum of 30%) were not in agreement. This Consultation Report was submitted to the Planning Inspectorate in December 2014 and a formal legal process was completed during a six-month period. All responses were reviewed and comments were categorised into principal topics. Changes to the design were agreed and based on environmental, cost and technical design considerations.

The earliest geophysical fieldwork took place in 2009. Archaeological investigation was carried out between 2016 and 2018 in accordance with the DCO Written Scheme of the Archaeological Investigations [40]. In July 2016 the client contracted the A14 Integrated Delivery Team for Phase 1 Site Investigations. A team of 250 archaeologists led by experts from MOLA Headland Infrastructure explored 33 sites across 360 hectares. More than 40 separate excavations were completed by summer 2018, uncovering around 25 settlements, dating from prehistoric to mediaeval periods with:

- 40 Roman industrial pottery kilns along Roman roads;
- Seven prehistoric burial grounds;
- Eight Iron Age to Roman supply farms, some with wells;
- Three prehistoric henge monuments;
- Two post-medieval brick kilns;
- One Roman distribution farm with military finds;
- Three Saxon settlement sites, one with royal connections;
- One deserted mediaeval village occupied from 8th to 12th century.

The community outreach Programme Mobile Visitor Centre of the project was set up and organised over 70 events. It included a Summer Community Dig, a programme of talks on UBH findings and guided tours. The news was published in village newsletters and in the Cambridge Archaeology page on Facebook. To meet the excavation demands, special attention was given to engage volunteers in archaeology. An accredited on-site training programme was set up in order to provide the participants with the skills to work on archaeological sites in the UK. As part of an Archaeological Outreach Programme, the team collaborated with local schools and provided opportunities for young people to gain work experience in archaeology.

## 4. Discussion

The five cases provide good examples of the role of digital and mobile technologies in engaging the community with UBH assets. The experiences have been analysed and the issues relevant to all cases identified, although they differ in the UBH typology, genesis

and history. The cases were discussed following the established common framework, and consider the three dimensions—the used digital technology, the means of organising the stakeholders' engagement and the obtained results from activating UBH assets. The main results are displayed in Table 1.

**Table 1.** Main characteristics of UBH activating—a brief comparison.

| Case | Kind of Underground Heritage | Digital and Mobile Technology | Means of Community Engagement | Results |
|---|---|---|---|---|
| *Matera Città Narrata—Tales of a City, Matera, Italy* | Rock-cut cave houses | Virtual reality panoramas, 3D reconstructions, applications, free website and mobile access to a vast amount of information for tourists and enables the stakeholders' engagement. | Storytelling; Free web, mobile and Wi-Fi access. | Local people tell their stories and visions in videos; A section in the platform for further stakeholders to share their own stories; Information on the heritage values available in different ways, formats and languages. |
| *Escoural Cave, Montemor-o-Novo, Portugal* | Caves with megalithic remains | 3D reconstructions through digital photogrammetry and laser scanning; Online and open-access platform, digital and open-access library, interactive maps. | Online platform (Morbase) created for/by locals; Collection of memories; Co-management of the UBH site. | Co-creation and dissemination of knowledge; Online and open access platform; Multisectoral management; Active role of the local community in tourism. |
| *Spanish Civil War, Valencia, Spain* | Military structures, such as air raid shelters, fortifications, trenches, bunkers, ditches, etc. | Online and open-access website; open inventory using GIS. | Community's feedback to inventory (cataloguing and reviewing information); Online forms for submission of non-inventoried heritage assets. | Extensive list of heritage elements; Community involvement in decision-making processes for heritage, memory and history recovery and conservation; Co-creation strategy. |
| *COMEVA(?) Project, Ticino, Switzerland* | Stone caves, extraction galleries | Official social network account; Parametric design tool. | Open-access platform to collect needs, visions, ideas, suggestions, and insights on the quality of public spaces around UBH (updated in real-time) | Map of ideas; Young people engaged in discussion forums; Attract online and offline users. |
| *A14 improvement scheme, Cambridge, UK* | Remains of 25 settlements, dating from prehistoric to mediaeval periods | Online forums for different stakeholders | Multi-staged non-statutory and statutory consultations; Non-statutory engagement included community digs, newsletters, social media. | Correct timing of consultations (before and after archaeological investigations) to safeguard the local environment and UBH; Online forums for different stakeholders, supplemented by offline events. |

From the cross-cases analysis, it is possible to distil four lessons concerning the relationships between UBH, people's engagement and the development of a smart community.

*4.1. First Lesson: The Value of Community and Local Stakeholder Involvement and Potentials of Digitalisation in Activating a UBH*

Community engagement is an essential component of creating a dynamic social environment that supports the activation of cultural heritage, including UBH. In addition, increasing engagement and citizen involvement broadens and helps the development of people's spatial skills. The cases illustrated different forms of community and stakeholders' involvement with UBH. It was noted that in all cases the emotional and identity-based connection with the heritage was achieved through civic participation and social accountability. As such, the actions raised community awareness in respect to UBH values, showing new meanings of UBH assets acquired over time and the possible changes in the landscape, noting that "Landscape is not inert. It is a social construct that is constantly being negotiated" [41]. Certainly, all these improve the quality of territories, and thus in turn, people's lives.

The Italian case exemplifies how successful cooperation between public and private stakeholders and local communities is able to increase the value of UBH, and how the multiple values of UBH can be preserved and communicated to future generations. The well-balanced research content of the *Matera Città Narrata* digital platform, smart involvement of the local community and accessibility is a good example of a smart performance of civic engagement. Giving citizens a stake in decision-making provides a better outcome, as the case of the Escoural Cave proves. The local community has been engaged with UBH since the cave's discovery and had an active role in the process of its safeguarding and dissemination. This case also clearly stands for the potentialities of UBH digitisation. The new digital tools allowed an integrated and rigorous study of such hidden UBH and, simultaneously, a wider dissemination among national or international scholars, as well as among a non-academic public. In this way, digitisation enables the discovery and a virtual appropriation of this heritage asset that may not always be accessible.

The working model adopted in Valencia, with a shared agenda and objectives, demonstrates the benefits of continuous active participation of local stakeholders, since projects which take a long time tend to show high rates of abandonment and lost interest from locals. The Spanish case shows that the network of local associations was a key factor in safeguarding social memory, creating a connection between social identity and historical references. The continuous dialogue and permanent cooperation between local administration and civil society, and the use of European funds, facilitated stable governance structures, indispensable for the processes of recovery and enhancement of UBH [42,43]. The Italian and Portuguese cases show that new technologies can be useful tools in the rediscovery, recovery and dissemination of intangible and tangible heritage values. They help to build cultural identity and a sense of belonging for the local community. Similarly, the Spanish case reveals that the community, through its collective memory and its knowledge of the history and territory, is an important element for mapping heritage assets and subsequently defining the best ways to preserve them.

The Swiss case underlines the community's attention to social and environmental aspects of the cross-border region. The case of the Gole della Breggia Park combines, in a virtuous way, public engagement and UBH topics where the reuse of the caves strengthens the sense of belonging of the local community with the potential for extending this successful model to the wider region. The UK case demonstrates the benefits of using combinations of different types of public engagement in UBH embedded in large infrastructure projects. It highlights that incorporating historical values into these projects helps promote sustainable change in planning, where the time management for statutory public consultations and site investigations and the curatorial role of archaeologists in any schemes are of high importance. In particular, it is suggested that a statutory increase of the length of public consultation to the design and construction stage could improve stakeholder's confidence in the successful project delivery. It is also noted that in large infrastructure projects where UBH and archaeology form only a small proportion of a wider project, the archaeological findings can also be used as a tool to engage the community.

*4.2. Second Lesson: Variegated Forms of Place-Based Engagement*

The five cases demonstrate different tools, forms and strategies used for public engagement. All of them feature a strong creative and sensitive process for place-based civic engagement. They include multistage statutory and non-statutory public consultations, formation of strategic stakeholder boards, creating inventories and using storytelling to enrich participation—not only from specific audiences (scientists, technicians, politicians and decision-makers), but also from local, general public. The community engagement, as the cases prove, leads to improved outcomes, where the community becomes more aware of its own culture and the new meanings of heritage, places, and landscapes.

The Spanish case offers evidence of different levels of stakeholders' interactions: (1) State–Regional, Regional–Provincial, Regional–Local, and Provincial–Local (those who own or have jurisdiction over heritage assets), in order to create regulatory and financial frameworks to promote the recovery of the identified heritage; (2) the network of associations enabled the creation of public–private alliances which promoted projects for heritage recovery and encouraged the involvement of the local communities; (3) citizens involved in the data collection, identification of assets, review and cross-checking of information collected. The Italian case also presents successful outcomes in terms of the engagement and involvement of stakeholders and the local community. On the one hand, the participation of public and private agencies at city, regional and national levels has brought a successful result for the valuation of the UBH. On the other hand, the involvement of the local community contributed to enriching the UBH assets through stories and visions, and these in turn, to the conservation and generation of the community's common values. The Swiss case shows innovative opportunities for the participation of different audiences, in particular younger generations, and helps to understand the needs of the new generation in relation to the local environment. The project also demonstrates how working offline helps to form an online community. These processes require time commitment and discussion opportunities in order to create a solid and lasting engagement.

The Portuguese case shows how the joint effort of multi-level stakeholders can promote new dynamics of knowledge creation at the local level and its dissemination to a global society. Memories are collected through the engagement of the local community, allowing the attachment of values intimately linked to the Escoural Cave. These memories and values are made available online through a dissemination platform, designed ad hoc by the municipality. Local people play a key role in the platform maintenance and management. In this way, locals are engaged in protecting, conserving and disseminating UBH sites.

The UK case confirms the benefits of multistage public consultations, which included both non-statutory and statutory consultations combined with various engagement activities. It was noted that the early engagement was about the benefits of the scheme and the process. Later engagement involved the discussion on what is going to happen, how people will be impacted, and the co-creation of common solutions. Statement of Community Consultation (SoCC) is a legal requirement in the UK and it proves to be a successful tool to create a strategy and framework for community engagement. The Strategic Stakeholder Board in the UK case was a good example of the body responsible for the public consultations which used different outreach activities that contributed to the legacy of the project, including forums for strategic stakeholders, community digs, engaging volunteers in archaeology, Mobile Visitor Centre, etc.

*4.3. Third Lesson: The Benefits of Digital Technology for Civic Engagement*

Digital tools potentially offer opportunities for enhancing UBH and activating community engagement, as they give and increase the visibility of an asset, which can be hidden and unknown [1]. The cases demonstrate that digital technologies were able to connect vertically the underground with the overground, and horizontally the local community/communities and different stakeholders, with various levels of influence and decision powers. Digital and mobile technologies, as the experiences show, can support the construction of scientific, technical and/or traditional "local situational knowledge" [44].

Digital technologies create new opportunities for the participation of different stakeholders by facilitating new networks and interactions [45]. They also help to expand knowledge and content production, and bring scientific knowledge production closer to the locals. In addition, they create new learning opportunities and increase citizen science alongside promoting more horizontal communication processes, which are broader and less formal.

Regarding the technology and innovation aspects, the Italian case, *Matera Città Narrata*, offers several ways of accessing a variety of content depending on the type of device used. The focus of the project was not on the new technologies, but on the improved accessibility and storytelling to connect people to the heritage. Nonetheless, it has to be noted that the more capability the user device has, the richer the forms and interactivity of the content that can be provided. In the case of the Escoural Cave, the smart approach allowed to combine the potential of digitisation, as a strategy for the dissemination and safeguarding of UBH, with the integration of the local community with UBH as a knowledge holder and owner of a local heritage.

It was also noted that the accessibility to a UBH asset, in particular for people with disabilities, can be partially solved by digital technologies by producing 3D virtual reconstructions to allow access for a larger number of visitors. As such, the website of the Spanish case was established as a dynamic, shared and open inventory, with an open feedback channel for the community and stakeholders. It facilitates the documentation through data collection, heritage mapping and knowledge of its status [46]. It also contains an open and accessible virtual informational space from both heritage and pedagogical perspectives. It can be used as a working tool as it provides accurate information for experts interested in carrying out studies on the topic.

The Swiss case also used the digital platform to address young people's needs and to develop their shared vision on public spaces. It investigated the successful practice in using social online platforms together with offline engagement activities and discovered that working offline helps to create an online community. The UK case used the same strategy of public engagement, where online forums for different stakeholders were supplemented by offline events.

### 4.4. Fourth Lesson: Co-Creation Process Provides Better Results in Community Engagement

The co-creation process was used as part of the public participation in all cases to collect views and produce different contents suitable for the project's needs. Storytelling by voices, using knowledge and authentic experiences of scholars, artists and children, was used in the Italian case to co-create the community spirit and to develop new content. At the same time, the group of experts followed the appropriation of UBH valorisation by the community and new innovative dimensions were added to the platform.

The case from Portugal shows that an intersectoral effort and excellent work developed by the multidisciplinary team allowed greater involvement of the community, not only through the provision of content, but also by being the content's producers and contributing to the reconstruction of collective memory that otherwise risked being lost. The Spanish experience shows that focusing first on establishing a legislative and regulatory framework, followed by efforts to raise awareness of these new regulations among the main stakeholders, generated local funding mechanisms and enabled them to identify other funding channels. Thanks to a well-coordinated stakeholder engagement and a flexible working model, adapted to the priorities and circumstances of the moment, successful efforts were made to create tools and strategies for heritage protection and consider other aspects linked to restoring war victims' dignity [43]. As such these actions re-established the connection of the population and tourists with the military heritage. In this sense, the recovery of ethnographic elements linked with the underground heritage is used as a tool to increase the confidence and identification of the population with the heritage to be recovered and enriches the heritage values of tangible cultural resources.

The Swiss case demonstrates the benefits of offline and online co-creation processes, the latter supported by social platforms and the website. A parametric representation was

used for online co-creation to evaluate design alternatives and their impacts on the local environment. This allowed the community to visualise and compare the initial situation and future outcomes, giving a realistic view of their choices. The results of the online co-creation served as the basis for offline discussions. In particular, they addressed the issues connected with the quality of public spaces and their relationships with the proposed constructions. In the UK case, a co-creation process was used in statutory and non-statutory consultations to collect opinions on the final design of the scheme, which were related to needs and wishes of stakeholders and in line with the Highways Agency's objectives. Questionnaires and online forums were used as a part of the co-creation process to receive feedback from the stakeholders on the proposals, which were reviewed by experts and designers, and responses were categorised into principal topics. Changes to the design were agreed and based on the comments of the stakeholders, on costs, and on environmental and technical aspects of the design.

## 5. Conclusions

The word 'smartness' is usually understood as a high degree of personal intelligence. It also describes the use of digital solutions to increase the operational efficiency of networks and services. In the framework of exploring valorisation strategies, smartness goes beyond the traditional definitions as it becomes evident that UBH recovery is part of the territorial development. With a clear strategy to obtain all relevant stakeholders on board, activating UBH assets bears a significant potential for generating economic activities. It is suggested that the smart valorisation strategies for UBH contain the combination of a strong leadership, a strategic stakeholder board, a plan for public engagement and the curatorial role of historians/archaeologists. It also uses a variety of consultation tools (e.g., co-creation, digital tools, etc.) to ensure that everyone has the opportunity to participate (as evidenced by the Italian and Portuguese cases). Digital and mobile technologies enable a wider participation in decision-making processes, as annotated by [45]. All this should be backed up with the legislative and regulatory framework, which was clearly demonstrated by the Spanish and UK cases. Multi-stage consultations in the UK case showed the importance of combining non-statutory route options and a statutory consultation with various engagement activities. Early engagement focuses on the benefits of the scheme and the process. Later engagement is about what is going to happen and how people will be impacted. Timing of the statutory consultation during, and most importantly after, site investigations, helps to avoid, minimise or offset harm to the local environment and UBH.

Leadership, as shown by the cases, is a critical determinant of success or failure. The five cases showed that different institutional levels are able to provide strong leadership. As such, in the Spanish and Swiss cases this was taken by the regional government; in the UK case by the local authorities, in the Italian a shared leadership was put in place, while the municipality assumed the leadership in the Portuguese case. In this framework, the strong leadership enabled the identification and cataloguing of the UBH assets and encouraged the dialogue between the different levels of administration (owners of the heritage) to create regulatory and financial instruments, and between local administration and the wider public to promote the heritage recovery. This practice was also used in the UK, where county councils led public consultations and established channels of communication so that all citizens can contribute to the recovery process and engage in the process of identification. The planning and design processes define roles for each agent involved and a common work agenda with shared objectives. The local networks are identified as the main stakeholders in the promotion of the recovery of historical (intangible) memory, which can be used as a tool to increase the confidence and participation of the local community with the heritage to be recovered and, as in the UK case, to engage the community to give an opinion on the issues, even those not related to UBH.

Digitisation is a successful strategy for the valorisation, wider dissemination and safeguarding of UBH, as shown by the Portuguese, Italian and Swiss cases. However,

online engagement should be mirrored by offline events, as evidenced by the successful experiences of the Swiss and UK cases. Digital solutions provide a virtual environment of open and accessible information for citizens, where everybody can contribute new or edit existing content. They facilitate a successful cooperation of all stakeholders from the public and private sectors and provide a smart monitoring tool for the community's engagement. Digital technologies also enable the discovery and virtual appropriation of the heritage asset, which may not always be accessible, and provide a 3D reconstruction to allow access to a larger number of visitors.

All the above is the endeavour of activating UBH into a community catalyser and preparing cities for a smarter future. As the five cases show, UBH creates a suitable framework to start thinking about smartness not only through the presence of smart digital experiences, but also from the perspective of social accountability and civic participation.

**Author Contributions:** Conceptualization, C.S.C., M.M., T.R., M.d.C.S.B. and M.A.E.; methodology, C.S.C. and R.V.; formal analysis, C.S.C., R.V., M.M., T.R., M.d.C.S.B. and M.A.E.; research, M.A.E., R.V., T.R., M.d.C.S.B. and A.R.; writing—original draft preparation, C.S.C., T.R., R.V., M.d.C.S.B., M.A.E. and A.R.; writing—review and editing, C.S.C., R.V. and T.R.; supervision, C.S.C. and R.V. All authors have read and agreed to the published version of the manuscript.

**Funding:** This research received no external funding.

**Data Availability Statement:** Not applicable.

**Acknowledgments:** This article has been prepared by the members of WG4 Planning Approaches of the COST Action CA18110 "Underground Built Heritage as Catalyser for Community Valorisation", https://www.cost.eu/actions/CA18110. The Escoural Cave study would not have been possible without the support of the executive bodies of the municipality of Montemor-o-Novo in the sharing of data and information.

**Conflicts of Interest:** The authors declare no conflict of interest.

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
