# Peer review of "Smart Thinking on Co-Creation and Engagement: Searchlight on Underground Built Heritage"

_smartcities, doi:10.3390/smartcities6010019_

Round 1

Reviewer 1 Report

Dear authors,

Thank you for you manuscript. Please find my comments as below. 

1. I could not see line numbers. Please correct this sentence in the Introduction for English "The interaction between the beneath ..." 

2. In the Introduction, I don't fully understand what is UBH and how are they utilized. Can you please provide a more explanation? Perhaps the first and second paragraphs can be rearranged. I see there is an example for UBH (vaccination center). 

3. The third Introduction paragragh does not bring any value. Can you add some literature examples for UBH which would show how UBH has been used so far, in what other ways is UBH defined, what is missing in literature and what new thing your paper brings? 

4. Materials and Methods can have subheadings - I would suggest to add Analysis as subheadings to section 2 as it looks quite strange otherwise. Please also rephrase the introduction to Materials and Methods so it has a more scientific look.  

5. The cases described have a report like format. Please change the style of writing so it focuses more on the 'engagement' and people. Currently I do not see the need to see numbers of videos and post. If you wish to mention the numbers, what is benefit to the community? As in, the overall outcome. 

6. I am missing a diagram to illustrate UBH and their use for the public and society. 

7. Stakeholder engagement is only mentioned in 3.5 A14. What about the others sites? 

8. Please highlight the challenge / profile for each site either in the text or table form. At the moment there is a lot of text that can be removed or refined. 

9. Discussion and Conclusion can be improved after the above sections are revised. 

10. Please also show results in a table form or another form. 

Author Response

  1. it is a quotation and should not be changed.
  2. A better explanation and few examples are added. 
  3. added - see 2).
  4. We prefer to keep this without subheadings. We are not sure what is meant by "more scientific look".
  5. Some cases are rephrased - but some specific information is kept as they represent the varieties of actions in the cases.
  6. 8. and 10. We do not see the need of diagrams/tables - the other 2 reviewers do not mention this
  7. Some cases are reworked in this issue.

Reviewer 2 Report

Well articulated article, clearly presented and well discussed messages. The cases chosen are appropriate and well presented, even not clear the grid of structuring them. The theoretic foundation can be extended, with more references to the actual debates, and a clearer link between ICT and the UBH.

From a methodologic point of view, indicators for structuring the study cases and for doing the comparative analysis would have been helpful. More hints about the comparative approach would have been needed. 

Author Response

Thank you very much for the review and suggestions. These are made directly in the new version of the manuscript!

Reviewer 3 Report

This is a very interesting article, well structured, making an important contribution to scholarship. The conclusions are presented in a clear and original way (in the form of lessons that can be a valuable inspiration for other localities and stakeholders engaging in a more effective use of the potential of underground built heritage).

The article could benefit from improving the following two aspects:

1. In the introduction of the article, it would have been useful to devote more attention to the state of research to date on the valorisation and exploitation of underground built heritage.

2. With regard to the experiences analysed (point 3), there is a certain imbalance in terms of the representation of the different sites. While locations 3.1., 3.2. and 3.5. are duly characterised in terms of the given type of underground built heritage and its geohistorical context, in the case of locations 3.3. and 3.4. it is more difficult for the reader to understand what type of sites are involved.  

Author Response

Thank you very much for the review and suggestions! We updated the issues raised in topics 1 and 2 directly in the new version of the manuscript!

Round 2

Reviewer 1 Report

Dear authors, 

Thank you for the revised version. 

I have some comments. 

1. It would be nice to see what was the research question in the Introduction. 

2. A table describing the five cities and a general description would still be good to add as there is a lot of text. 

Author Response

Thank you again for your review and ideas!

We tried to make the research question more clear -  see page 3.

and added a table with the most relevant data from the cases - this helps to make clearer the contribution of each case to the debate - see page  11 ff - the text in blue are new or adapted to introduce the table.